# Dialkyl Carbamoyl Chloride–Coated Dressing Prevents Macrophage and Fibroblast Stimulation via Control of Bacterial Growth: An In Vitro Assay

**DOI:** 10.3390/microorganisms10091825

**Published:** 2022-09-13

**Authors:** Silvestre Ortega-Peña, Mario Chopin-Doroteo, Alberto Tejeda-Fernández de Lara, David M. Giraldo-Gómez, Rosa M. Salgado, Edgar Krötzsch

**Affiliations:** 1Laboratory of Connective Tissue, Centro Nacional de Investigación y Atención de Quemados, Instituto Nacional de Rehabilitación, “Luis Guillermo Ibarra Ibarra”, Mexico City 14389, Mexico; 2Microscopy Core Facility, School of Medicine, Universidad Nacional Autónoma de México (UNAM), Mexico City 04510, Mexico

**Keywords:** hydrophobic, bacterial growth, inflammation, cytokine, fibroblast, macrophage

## Abstract

In this work, we evaluated the direct effect of a dialkyl carbamoyl chloride (DACC)-coated dressing on *Staphylococcus aureus* adhesion and growth in vitro, as well as the indirect effect of the dressing on fibroblast and macrophage activity. *S. aureus* cultures were treated with the dressing or gauze in Müller-Hinton medium or serum-supplemented Dulbecco’s modified Eagle medium. Bacterial growth and attachment were assessed through colony-forming units (CFU) and residual biomass analyses. Fibroblast and macrophage co-cultures were stimulated with filtered supernatants from the bacterial cultures treated with the DACC-coated dressing, following which tumor necrosis factor (TNF)-α/transforming growth factor (TGF)-β1 expression and gelatinolytic activity were assessed by enzyme-linked immunosorbent assays (ELISA) and zymography, respectively. The DACC-coated dressing bound 1.8–6.1% of all of the bacteria in the culture. Dressing-treated cultures presented biofilm formation in the dressing (enabling mechanical removal), with limited formation outside of it (*p* < 0.001). Filtered supernatants of bacterial cultures treated with the DACC-coated dressing did not over-stimulate TNF-α or TGF-β1 expression (*p* < 0.001) or increase gelatinolytic activity in eukaryotic cells, suggesting that bacterial cell integrity was maintained. Based on the above data, wound caregivers should consider the use of hydrophobic dressings as a first option for the management of acute or chronic wounds.

## 1. Introduction

The presence of microorganisms is the most common cause of delayed wound repair. The amount, growth, and virulence of hosted microorganisms determine the wound state and prognosis [1]. The release of antimicrobial compounds and phagocytosis by host cells is required for infection control [2]. However, microorganism endotoxins are also released during the removal of an infectious agent, which can exacerbate the inflammatory response and contribute to the development of a chronic condition [3]. The control of microorganism growth through microbial lysis does not prevent residual damage, which exacerbates the host immune response. For this reason, the aim of infection treatment should be focused on limiting or preventing microbial growth as early as possible [4]. Some non-microbiolytic strategies for the improvement of wound repair involve the removal of microbes from the wound bed without lysis. Negative-pressure wound therapy mainly focuses on wound bed preparation, where exudate removal contributes substantially to reduction in the microbial load in the wound after such treatment [5]. The use of non-bacteriolytic antibiotics [6] or dressings with chemical or biochemical microorganism-binding surfaces [7] has been considered an alternative approach for the treatment of acute and chronic wounds. Advanced wound-care involves the use of materials that, among other properties, can retain wound moisture and limit microbial growth [8]. One such material is an acetate fabric dressing impregnated with dialkyl carbamoyl chloride (DACC), a hydrophobic fatty-acid derivative [9]. The physicochemical properties of this material allow it to attach bacteria through the hydrophobic elements on their cell surfaces. Removal of the dressing can clear a considerable proportion of bacteria (e.g., *Pseudomonas* and *Staphylococcus* species) [10] and/or biofilm from the wound bed, reducing the bacterial load in the moist wound environment [11]. The use of such dressings is considered to be a primary option for surgical-site infection prevention [12] and for the management of different types of contaminated or infected injuries [13], with the secondary advantage being that DACC promotes fibroblast proliferation, which is favorable for wound repair [14].

*Staphylococcus aureus* (*S. aureus*) is a bacterium with a typical hydrophobic surface, mainly in its stationary (rather than exponential) growth phase. This hydrophobicity is attributed to the different molecules associated with proteins anchored to the bacterial cell surface in the cell membrane and/or wall [15]. The properties of *S. aureus* strains with highly hydrophobic surfaces never change, regardless of the nutritional or polar conditions [16]. Thus, *S. aureus* (the bacteria found most frequently in wounds) can be removed from wounds through the use of DACC-coated dressings. Although many studies have examined the effectiveness of DACC-coated dressings for wound repair [7,12,17], our goal was to determine the ability of the DACC-coated dressing to reduce the growth of *S. aureus* when cultured in different media and under different aerobic conditions. We also focused on exploring how the supernatants from *S. aureus* cultures treated with the DACC-coated dressing could affect the expression of pro-inflammatory cytokines and extracellular matrix remodeling, when added to fibroblast and macrophage co-cultures.

## 2. Materials and Methods

### 2.1. Microbiological Assays

We performed an in vitro assay to examine the attachment and growth of the oxacillin-sensitive *S. aureus* strain ATCC29213 in the presence of an acetate fabric dressing coated with DACC (Cutimed^®^ Sorbact^®^; Essity BSN Medical GmbH, Hamburg, Germany). First, bacteria were grown under aerobic conditions on 5% sheep blood agar for 24 h at 37 °C. We prepared a 0.5 McFarland standard bacterial suspension (1.5 × 10^8^ colony-forming units [CFU]/mL), using the Clinical and Laboratory Standards Institute protocol [18], and performed serial dilutions in Müller-Hinton medium (Becton Dickinson and Company, Sparks, MD, USA) or Dulbecco’s modified Eagle medium (DMEM; GibcoTM, Life Technologies, Brooklyn, NY, USA) supplemented with 10% fetal bovine serum (GibcoTM) and 2 mM glutamine (GibcoTM). A piece of DACC-coated dressing or cotton gauze (1.4 cm diameter) was placed in each well of a 24-well culture plate (Costar; Corning Inc., Corning, NY, USA), and 0.5 mL of *S. aureus* suspension, equivalent to 1.5 × 10^6^ CFU (in order to mimic the bacterial presence in infected tissue) was added. The dressings were immersed mechanically in the *S. aureus* cultures and rotated every 30 min to ensure that they were completely embedded in the media. Then, the dressings were incubated for 3 h with gentle orbital shaking. Wells without dressings, but with bacterial suspension with or without oxacillin (10.13 μg/mL; Sigma-Aldrich, St. Louis, MO, USA), were used as controls. *S. aureus* cultures in Müller-Hinton medium were incubated aerobically, while those with supplemented DMEM were grown under microaerophilic conditions in the presence of 5% CO_2_, in order to simulate the wound environment [19].

The dressings were then collected and washed twice with 1 mL phosphate-buffered saline (PBS) solution. The washes and the corresponding supernatants were pooled in 15-mL polypropylene tubes. The empty wells were washed twice with 0.5 mL PBS, and the washes and supernatants were pooled. From each bacterial suspension, 100 μL were diluted with 9.9 mL PBS. After stirring, 20 μL of the final suspension were deposited in a Petri dish, and warm trypticase soy agar (Dibico SA de CV, Estado de México, Mexico) was added. When the agar had solidified, the dishes were incubated for 24 h at 37 °C, and the CFUs from the bacterial supernatants were counted.

To count the bacteria adhering to or trapped in the dressings, each dressing was washed with 1 mL PBS/0.5% Tween 20 (Sigma-Aldrich) for 1 min with vigorous stirring. Each wash was diluted with 9 mL PBS, and the CFUs in 20 μL of the final suspension were counted as described above. Previously washed wells were left to dry for 30 min at room temperature, and 0.5 mL of 0.1% violet crystal solution (Sigma-Aldrich) was added for 15 min to stain the residual biomass. The stain was then removed, and the wells were washed three times with 0.5 mL PBS. The plates were allowed to dry again, and the stain was extracted with 0.5 mL 90% ethanol. Then, 200 μL of each solution was quantitated using a colorimetric method with an xMark™ microplate absorbance reader (Bio-Rad Laboratories, Inc., Hercules, CA, USA) and a 490 nm filter [4]. 

### 2.2. Stimulation of Eukaryotic Cells with Supernatants of S. aureus Cultures

Murine macrophage cultures (ATCC cell line RAW 264.7) were maintained in supplemented DMEM with antibiotics (100 U/mL penicillin and 100 µg/mL streptomycin, Gibco) at 37 °C with 5% CO_2_. When confluent, 1 × 10^5^ cells per well were deposited in 24-well culture plates with 1 mL supplemented DMEM with antibiotics. The plates were incubated for 24 h at 37 °C with 5% CO_2_. The media were removed and the cultures were treated with 45 μL filtered supernatant (applied using a 28-mm syringe filter, Microcon SFCA membrane; Corning) from *S. aureus* cultures incubated with or without dressings in Müller-Hinton broth and 955 μL fresh supplemented DMEM with antibiotics. The plates were incubated for 24 h as previously described, and the conditioned media were collected and frozen at −70 °C until further analysis of tumor necrosis factor (TNF)-α protein expression.

We prepared macrophage and fibroblast (murine, 3T3) co-cultures. Eukaryotic cells were grown as described previously, where 9.5 × 10^4^ fibroblasts were deposited in 24-well culture plates with 250 μL supplemented DMEM with antibiotics. Immediately, cell culture inserts (Millicell, 12 mm, 0.4 µm polycarbonate; Millipore, Burlington, MA, USA) were placed and 1.13 × 10^5^ macrophages in 250 μL supplemented DMEM with antibiotics were added to the inserts. The co-cultures were maintained for 24 h at 37 °C with 5% CO_2_. Then, the culture media were removed carefully in order to avoid monolayer scratching, and the co-cultures were treated with a mixture of 125 μL fresh supplemented DMEM with antibiotics and 125 μL filtered supernatant from *S. aureus* cultures incubated in supplemented DMEM with and without dressings. The co-cultures were incubated as previously described, and the conditioned media were collected and frozen for the analysis of TNF-α and transforming growth factor (TGF)-β1 protein expression, as well as gelatinolytic activity.

### 2.3. Cytotoxicity Assay

To evaluate the cytotoxicity of eukaryotic cultures, due to their growth in the presence of the filtered supernatants, we prepared and treated macrophage cultures and fibroblast/macrophage co-cultures as described previously. After the final 24 h incubation, we added 25 μL of a 5 µg/mL solution of 3-(4,5-dimethylthiazol-2-yl)-2,5-diphenyltetrazolium bromide (MTT; Sigma). After 3 h, the media were removed and each monolayer was washed three times with PBS. The MTT reduction was assessed after dissolution of the formazan crystals with 200 μL dimethyl sulfoxide/isopropanol and colorimetric measurement of the solution at 570 nm [20].

### 2.4. Cytokine Expression Analyses

TNF-α and TGF-β1 protein expressions were quantified by enzyme-linked immunosorbent assay (ELISA) using Quantikine mouse TNF-α and TGF-β1 Kits (R&D Systems, Minneapolis, MN, USA), according to the manufacturer’s instructions, with 1:5 dilutions of the eukaryotic culture-conditioned media. Blank samples were prepared using supplemented DMEM with antibiotics, incubated in empty wells for 48 h. The cytokine values were normalized using MTT assay data.

### 2.5. Zymographic Evaluation of Gelatinase Activity in Eukaryotic Conditioned Media

To examine gelatinolytic activity in the conditioned media from eukaryotic cultures, polyacrylamide/sodium dodecyl sulfate 1% gelatin gel electrophoresis was performed, using protein content-standardized samples [21]. Gelatinolytic activity was semi-quantified by densitometry, using the Quantity One 4.6.3 software (Bio-Rad Laboratories, Inc., Hercules, CA, USA).

### 2.6. Microscopic Analysis of DACC-Coated Dressings

Dressing samples obtained separately from the experiments performed for CFU quantitation were washed once with PBS and fixed with 2% glutaraldehyde solution. Each sample was divided, and one portion was dehydrated with an alcohol gradient (50–100% ethanol) and xylene, then embedded in paraffin. Five micron-thick transversal sections were stained with hematoxylin and eosin, in order to identify attached bacteria, and photomicrographs were acquired at 630 × magnification under an Axio Imager Z.1 microscope (Carl Zeiss, Göttingen, Germany) fitted with a high-speed camera (AxioCam; Carl Zeiss, Jena, Germany) using the ZEN lite software V. 3.0 (Carl Zeiss, Jena, Germany). The other portions of the dressing samples were fixed in a sample holder with double-sided carbon tape and colloidal silver, air-dried for 24 h, and gold-sputtered (108A auto sputter coater; Agar Scientific, Essex, UK) for 40 s at 40 mA. Images were obtained by field-emission scanning electron microscopy (Crossbeam 550; Carl Zeiss, Jena, Germany) at 2 kV with a secondary electron detector [22]. Photomicrographs were acquired at low and high magnification.

### 2.7. Statistical Analyses

All experiments were performed in triplicate. CFU data from three independent experiments and eukaryotic cell data from two independent experiments were used. The results were analyzed using the GraphPad InStat 3.0 software (GraphPad Software, Inc., La Jolla, CA, USA). Quantitative variations among study groups were analyzed by different statistical methods, according to their normality and standard deviation differences. To determine the statistical significance, the number of CFUs in dressings was compared among the study groups using an unpaired t-test, and the numbers of CFUs in supernatants from bacterial cultures grown in Müller-Hinton medium were compared using the Friedman test with Dunn’s multiple-comparisons test. CFUs in supplemented DMEM, residual biomass, and TGF-β1 expression in macrophage and fibroblast co-cultures were assessed using one-way analysis of variance with the Tukey–Kramer multiple-comparisons test. The Kruskal-Wallis test with Dunn’s multiple comparisons test was used to compare TNF-α expression among groups of cultures and co-cultures. *p* values ≤ 0.05 were considered to indicate statistical significance.

## 3. Results

### 3.1. S. aureus Colony Arrangement Varies according to the Physicochemical Properties of Every Dressing

Bright-field microscopy of the fibers from the DACC-coated dressing or gauze showed abundant bacteria attached. Photomicrographs show single or small groups of bacteria adhering irregularly to the surface of the dressing (arrowheads in Figure 1a) and dispersed or aggregated cells at different depths in the gauze (arrow in Figure 1b). The DACC-coated dressing exhibited a wide distribution of bacteria, cotton gauze seemed to concentrate them in the middle portion of the fiber (Figure 1a,b). Typical *S. aureus* morphology and bacterial extracellular matrix deposition was confirmed by scanning electron microscopy (SEM) at high magnification (Appendix A). Furthermore, the pictures obtained by SEM showed that the DACC-coated dressing fiber presented a regular monofilament thread, opposite to cotton fiber, which was composed by several fine threads such as a rope (Figure 1c,d). After 3 h of culturing, *S. aureus* formed stable biofilms with an evident extracellular matrix in the DACC-coated dressing (Figure 1e,g); while planktonic cells and small colonies were observed among the cotton gauze fibers (Figure 1f,h).

### 3.2. A Lower Proportion of S. aureus in Cultures Treated with DACC-Coated Dressing Was Associated with Bacterial Retention

Up to two orders of magnitude more CFUs were present in samples cultured with dressings and incubated in Müller-Hinton medium than in those incubated in supplemented DMEM. The difference in bacterial retention between the dressings was only significant (*p* < 0.0001) for cultures prepared in supplemented DMEM, with gauze retaining 30% more bacteria than the DACC-coated dressing (Figure 2a). Control cultures prepared in Müller-Hinton medium exhibited less growth than gauze (*p* < 0.01). However, cultures treated with the DACC-coated dressing in supplemented DMEM showed significantly less bacterial growth than gauze and control samples (*p* < 0.001); except for the oxacillin group, which showed the least *S. aureus* growth in any medium (*p* < 0.01 and 0.001, when compared with cultures treated with the DACC-coated dressing or control and gauze treated, respectively; Figure 2b). Although all of the experimental groups showed less residual biomass than control (*p* < 0.001), the least residual biomass among the experimental groups was observed for the DACC-coated dressing and oxacillin samples (*p* < 0.01; Figure 2c).

### 3.3. Supernatants of S. aureus Cultures Incubated with the DACC-Coated Dressing Down-Modulated Inflammation Related Cytokine Overexpression and Diminished Gelatinase Activity

To determine whether the DACC-coated dressing could hold bacteria without lysis, we evaluated the effects of 0.22 µm filtrates of supernatants from *S. aureus* incubated with the dressing on macrophage cultures and fibroblast/macrophage co-cultures. Bacteria grew in Müller-Hinton medium or supplemented DMEM did not present altered eukaryotic cell growth (Appendix A); however, significantly reduced expression of TNF-α and TGF-β1 (Figure 3) and a proportional diminution of 92 kDa gelatinase activity (Figure 4) were observed in the eukaryotic cultures and co-cultures treated with the filtered supernatants of bacteria treated with DACC-coated dressing, when compared with the oxacillin-treated group. Those treated with gauze only presented reduced TNF-α expression.

## 4. Discussion

While many researchers have studied how to prevent or avoid hydrophobic interactions between bacteria and surfaces (e.g., biofilms), others have taken advantage of these phenomena to design a dressing that enables the removal of bacteria [23]. In this work, we demonstrated that *S. aureus* adhered to the considered DACC-coated dressing, while cotton gauze only retained the bacteria by simple trapping within its fiber structure. This effect importantly modified bacterial metabolism, promoting indirect effects on fibroblast and macrophage cytokine expression and gelatinolytic activity. 

Substrate-bacterial adhesion begins close to the surface of the material (at approximate depths of 50–100 nm) [23]. The adherence of *S. aureus* to surfaces is attributable mostly to hydrophobic macromolecules, whereas other binding mechanisms are hydrophilic. Although, in general, hydrophilic binding is physically stronger than hydrophobic bonding, the abundance of the latter exerts a stronger final effect [24]. The attachment of a bacterium to a hydrophobic surface is considered to be irreversible, as the water displacement between the bacterium and the solid surface is stabilized by hydrogen bonds. Bacteria bound to hydrophobic surfaces cease nuclease production, avoiding extracellular DNA degradation and, consequently, stabilizing and promoting the spread of biofilm [25]. For the DACC-coated dressing, this favorably reduced nuclease expression, which has been associated with the reduction in bacterial virulence [25]. Although gauze also traps bacteria, it does so in a mainly hydrophilic environment, which stimulates greater bacterial virulence. The use of hydrophobic dressings could lead to remove precisely the most dangerous forms of bacteria, as hydrophobicity in the cell surface has been associated with a linear increase in antimicrobial resistance [26].

In addition, we demonstrated that the binding of the bacteria to the DACC-coated dressing diminished bacterial growth, as to calculated number of CFUs in supernatants and dressings were lower than those observed in the gauze-treated cultures, as the gauze promoted uncontrolled bacterial growth. This phenomenon is a key factor in wound management, as gauze has been considered as a primary dressing for many years; despite its low cost and availability, the use of a cotton dressing may contribute to slowed wound healing and could promote wound trauma, as the gauze frequently adheres to the wound bed. When it is removed, the granulation tissue growing underneath is detached, leading to injury perpetuation [27]. In contrast to gauze, synthetic, non-adherent dressings have been considered for wound management as their fibrillar structures can protect the injury or even release antiseptics or wound-healing promoters without any tissue adherence; however, wound exudate management is a critical step when synthetic non-absorbent dressings are used [28]. 

Another important consideration regarding wound dressings is their “capability” to attach and remove biofilms. Mere clinical observation or inconclusive assays have suggested that dressings can detach biofilms from the wound; however, these data come from experiments where the authors observed only the material adhered to the dressing and did not consider the residual biomass left in the treated surface. For example, it has been reported that the DACC-coated dressing can bind biofilm structures of *Pseudomonas aeruginosa* and *S. aureus* [11], as in vitro assays evidenced by SEM have demonstrated that, after different lengths of incubation of the dressing with previously formed biofilms, the DACC-coated dressing exhibited progressive coverage with biofilm. Those results were inconclusive as, despite the fact that it was evident that biofilm formed on the dressing, the incubation time was sufficient to allow for superficial preformed biofilm bacteria detachment, followed by progressive attachment and maturation on the dressing, such as the results obtained herein, where planktonic *S. aureus* cultures formed biofilm structures on both, the well bottom and the dressing, after 3 h of incubation with the DACC-coated dressing. DACC-coated dressings can remove 0.7–2.9 × 10^6^ CFU/cm^2^
*S. aureus* of various strains, including those that are methicillin-resistant [29]. We obtained similar results for cultures of *S. aureus* grown in Müller-Hinton broth, but lower values for those grown in supplemented DMEM. In this work, we observed that the removal method for axenic *S. aureus* cultures affected TGF-β1 and TNF-α expression and gelatinolytic activity in fibroblast and macrophage co-cultures. Filtered supernatants from bacterial cultures treated with oxacillin and exposed to fibroblast and macrophage co-cultures showed increased TNF-α expression, as well as 72 and 92 kDa gelatinolytic activity, in agreement with previous findings [30]. Macrophage exposure of *S. aureus* lysates from beta lactam antibiotic treatment increased the release of inflammatory mediators, such as TNF-α and nitric oxide synthase [31]. Gram-positive bacteria treated with beta lactam antibiotics (e.g., flucloxacillin) have been reported to promote cell lysis and the release of antigenic molecules as a consequence of the inhibition of cell wall synthesis [32]. Peptidoglycan and lipoteichoic acid are two main components of the cell wall that can be released spontaneously into the culture medium during bacterial growth, and exposure to these molecules is increased in the presence of antibiotics. In addition, peptidoglycan and teichoic acids stimulate pro-inflammatory cytokine expression in peripheral blood mononuclear cells [32].

In addition, neutrophil cultures treated with supernatants of *S. aureus* cultures increased phagocytic activity and interleukin-8 secretion [33]. On the other hand, bovine mammary fibroblast cultures treated with lysates from heat-inactivated *S. aureus* cultures exhibited increased mRNA levels of pro-fibrogenic mediators, such as TGF-β1 and basic fibroblast growth factor [34], as well as increased matrix metalloproteinase (MMP)-1, -2, -3, -9 and -13 expression [35]. Such over-production of inflammation mediators can impair wound healing and generate tissue damage and septic shock. In the present work, filtered supernatants from *S. aureus* cultures treated with the DACC-coated dressing showed reduced TGF-β1 and TNF-α expression and 92 kDa gelatinase activity (which is related to the MMP-9 isoform) in fibroblast and macrophage co-cultures, suggesting that the physical removal of bacteria has advantages over their killing by antibiotic or antiseptic means, due to the control of excessive local inflammation and remodeling [6]. We also found that treatment with the filtered supernatants of bacterial cultures exposed to the hydrophobic dressing had no effect on eukaryotic cell growth, despite Falk and Ivarsson’s [14] observation of increased proliferation and migration in fibroblast cultures treated directly with DACC-coated dressing in an in vitro model of damage. These findings suggest that the dressing has additional properties which improve wound closure. However, it is necessary to be cautious about simultaneous treatments for wounds, as some proteases employed for wound debridement could break the fixing protein chains, preventing cell wall attachment to the hydrophobic surface of the dressing [23]. The bacterial growth medium did not affect the properties of the DACC-coated dressing in this study. In 1987, Mamo et al. [16] demonstrated that the physicochemical properties of the cell surfaces of auto-aggregating or hydrophobic *S. aureus* strains did not change significantly when grown under different nutrient conditions. This property is advantageous for treatment with hydrophobic dressings, as it means that differences in the wound exudate among patients would not affect the affinity of the bacteria for the dressing. The DACC-coated dressing examined in this study should be most effective on moist or exudative wounds, as dry lesions do not have the physicochemical conditions required for the interaction of microbiota with the dressing; its use on this type of wound requires the application of hydrating strategies, such as the use of hydrocolloids. For these reasons, mechanistic clinical trials are needed to demonstrate in vivo dressing effects, beyond simple visual or microbiological analyses. Different authors have presented alternative methods that contribute to preventing antibiotic and/or antiseptic microbial resistance (e.g., hydrophobic dressings) for wound repair [36].

## 5. Conclusions

The use of a DACC-coated hydrophobic dressing enabled the physical removal of *S. aureus* along with the maintenance of bacterial structural integrity. This hypothesis was supported by data obtained from the treatment of fibroblasts and macrophages with supernatants from *S. aureus* cultures treated with the DACC-coated dressing. Namely, eukaryotic cells expressed lower levels of cytokines than the controls. Moreover, the DACC-coated dressing demonstrated bacteriostatic effects which could contribute to bacterial quiescence, thus limiting the release of prokaryotic molecules that could stimulate inflammation. Clinical approaches can be derived from the data presented in this work, where prophylactic or therapeutic applications of hydrophobic dressings can be recognized beyond simply bacterial attachment and removal. Furthermore, in this work, we evidenced the relevance of integral bacterial removal, which is frequently forgotten in daily practice, as cotton gauze is still the most commonly used “inert” dressing. Perhaps it is time to recognize the benefits provided by other kinds of wound dressings over gauze, mainly from those recommended for dry or low-exudative wounds, such as polyvinylpyrrolidone, cellulose acetate, rayon, polyester, or other synthetic fibers. The information reported here could also be useful for furthering our understanding of prokaryotic–eukaryotic signaling during wound repair.

## Figures and Tables

**Figure 1 microorganisms-10-01825-f001:**
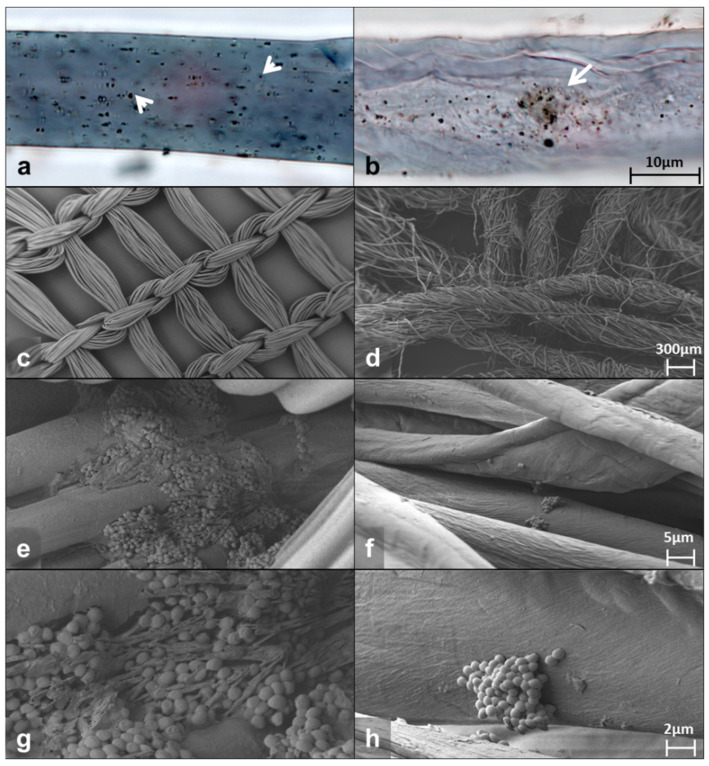
Representative photomicrographs of the DACC-coated dressing (left panels) and gauze (right panels) cultured with *S. aureus* in supplemented DMEM. Bright-field images (**a**,**b**); SEM images (**c**–**h**).

**Figure 2 microorganisms-10-01825-f002:**
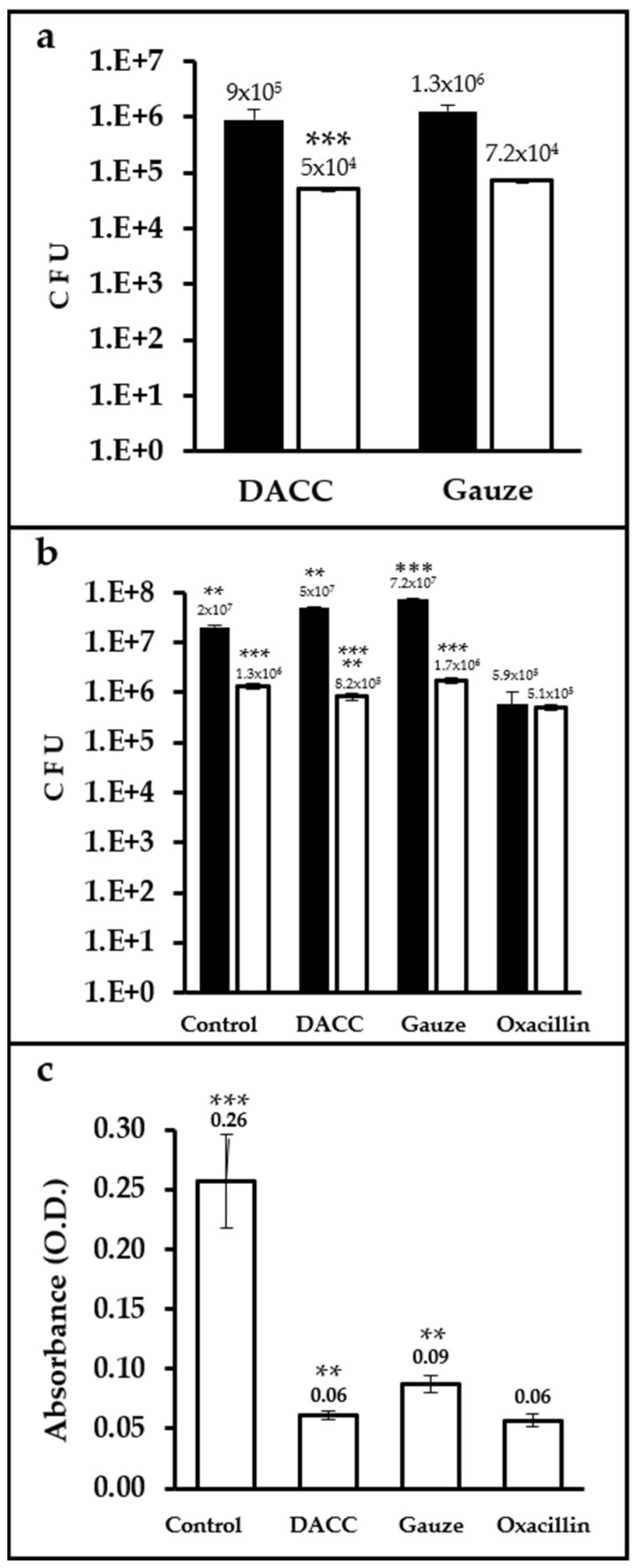
Quantification of CFUs and residual biomass in *S. aureus* cultures under different treatments: (**a**) Logarithms of mean CFUs of *S. aureus* retained in the dressings. The bacteria were cultured in Müller-Hinton broth (black bars) or supplemented DMEM (white bars); ***, *p* < 0.001 DACC vs. gauze, cultured in DMEM. (**b**) Logarithms of mean CFUs of *S. aureus* in the supernatants of cultures; **, *p* < 0.01 control vs. gauze, DACC vs. oxacillin; ***, *p* < 0.001 gauze vs. oxacillin, cultured in Müller-Hinton; **, *p* < 0.01 DACC vs. oxacillin; ***, *p* < 0.001 control vs. DACC, gauze, and oxacillin; DACC vs. gauze; and gauze vs. oxacillin, cultured in DMEM. (**c**) Mean of the residual biomass in the bottom of the well from the *S. aureus* cultures grown in DMEM. **, *p* < 0.01 DACC vs. gauze, gauze vs. oxacillin; ***, *p* < 0.001 control vs. DACC, gauze, and oxacillin. Error bars represent standard deviations.

**Figure 3 microorganisms-10-01825-f003:**
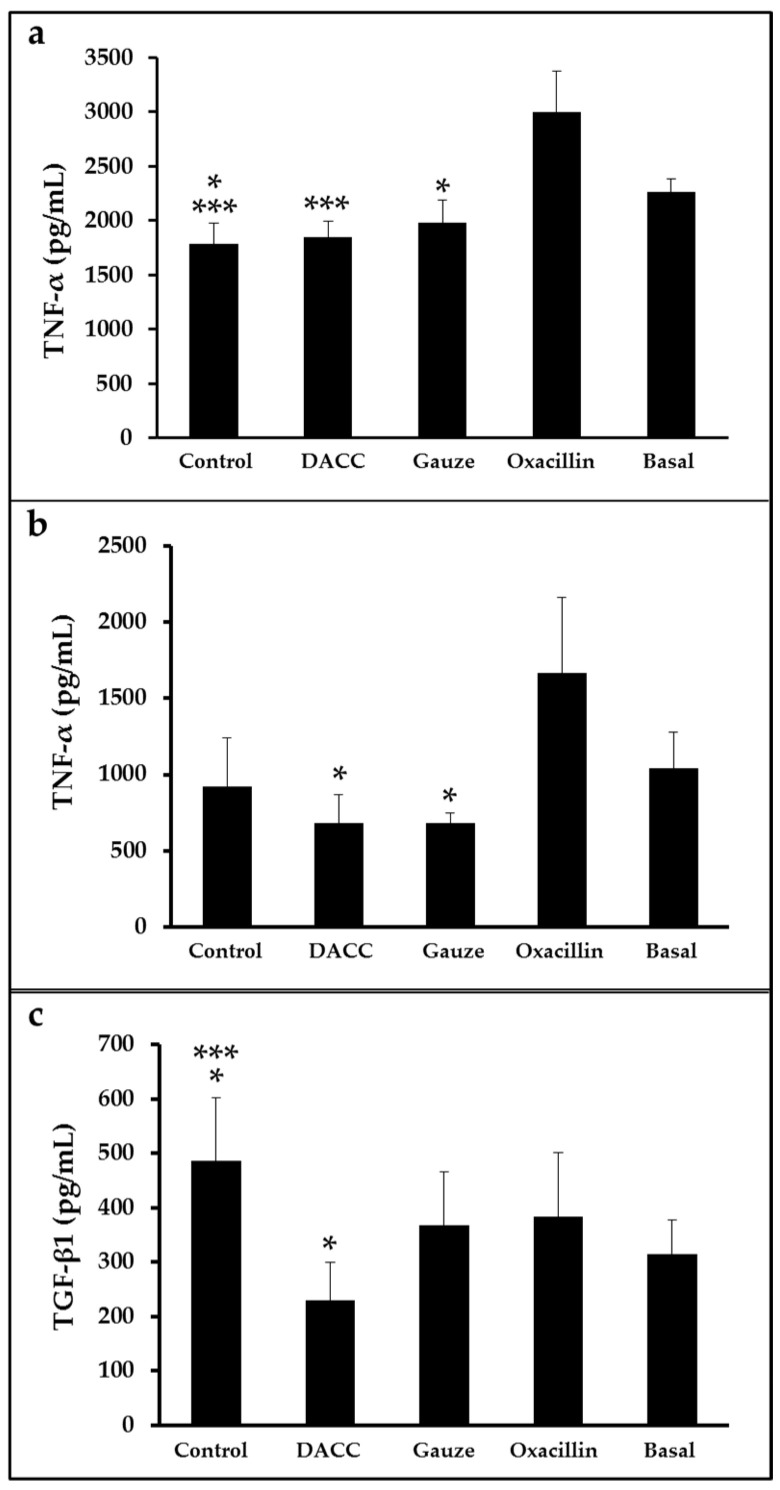
Expression of TNF-α and TGF-β1 in eukaryotic cell cultures: (**a**) Macrophages treated with the filtered supernatants of *S. aureus* cultured in Müller-Hinton broth. *, *p* < 0.05 control vs. basal and gauze vs. oxacillin; ***, *p* < 0.001 control vs. oxacillin and DACC vs. oxacillin. (**b**,**c**) co-cultures of fibroblasts and macrophages treated with the filtered supernatants of *S. aureus* cultured in DMEM. In (**b**), *, *p* < 0.05 gauze vs. oxacillin and DACC vs. oxacillin. In (**c**), *, *p* < 0.05 control vs. basal and DACC vs. oxacillin; ***, *p* < 0.001 control vs. DACC. Values are means ± standard deviations. The basal term refers to cytokine expression in the conditioned medium from non-unstimulated eukaryotic cultures.

**Figure 4 microorganisms-10-01825-f004:**
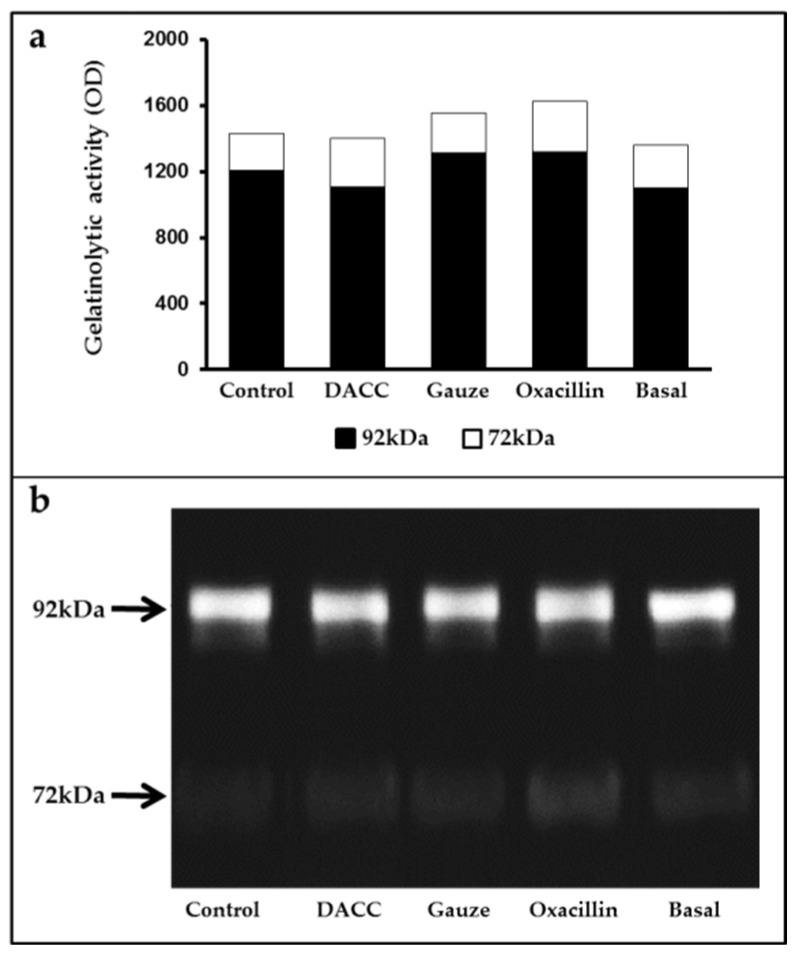
Average gelatinase activity derived from two independent experiments: (**a**) Black bars correspond to 92 kDa and white bars to 72 kDa species. The basal term refers to the conditioned medium from non-unstimulated eukaryotic cultures; (**b**) Gelatinolytic activity in the supernatants of fibroblast and macrophage co-cultures, as determined by gelatin zymography and polyacrylamide gel electrophoresis.

## Data Availability

Not applicable.

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
