# Peer review of "Dialkyl Carbamoyl Chloride–Coated Dressing Prevents Macrophage and Fibroblast Stimulation via Control of Bacterial Growth: An In Vitro Assay"

_microorganisms, 2022, doi:10.3390/microorganisms10091825_

Round 1
Reviewer 1 Report
The topic submitted is novel and adds significant research data to the existing field of microbial research. The article is not very well articulated and needs English language revisions and even formatting of the manuscript as per the MDPI guidelines. The manuscript needs to be checked for statistical significance and indicated with **, especially in fig 3,4. The introduction needs to be concise. Abstract and conclusion should include a sentence proposing the future direction of the present research. Also the commercial aspects of how pharma or biotech industries can benefit. Most of the methodology lacks citing the references from where the technique was adopted and used to conduct the study. Below are a few suggestions the authors are requested to discuss and cite in appropriate sections. Can be accepted after minor revisions.
Journal of Wound Care 25, no. 2 (2016): 76-82.
European Polymer Journal 136 (2020): 109919.
Wound Medicine, 7, 14-17.
Journal of Engineered Fibers and Fabrics, 10(4), p.155892501501000411.
Journal of Biomedical Materials Research Part B: Applied Biomaterials 108, no. 8 (2020): 3084-3098.
In Alginates in Drug Delivery, pp. 323-358. Academic Press, 2020.
BJS open, 4(2), pp.225-231.
International Journal of Biological Macromolecules 165 (2020): 1924-1938.
Author Response
Microorganisms Journal
Reviewer 1
Dear Reviewer 1:
Please find the revised version of the manuscript entitled “Dialkyl carbamoyl chloride–coated dressing prevents macrophage and fibroblast stimulation via control of bacterial growth: An in vitro assay”. Manuscript ID: microorganisms-1809877.
In this new version of the work we have significantly improved the manuscript. We have included one supplementary material, and we have modified some figures in order to be more explicit. Awkward sentences have been revised and corrected and the manuscript has been reviewed by the language editing office of MDPI; we have attached the English-editing certificate-47892 with this manuscript submission. In general, we consider that reviewers’ comments helped us to improve the work in a final clear and purposeful version. Below you can find a point by point response, where every change of the manuscript is tracked according to the corresponding line(s). Since the manuscript has been rewritten we have not tracked changes on it.
Respectfully,
The correspondence author.
Reviewer 1:
- Reviewer 1 comment: The article is not very well articulated and needs English language revisions and even formatting of the manuscript as per the MDPI guidelines.
Author response: According to Reviewer 1, in the new version of the manuscript we have improved the manuscript according to MDPI guidelines and it has been reviewed by the language editing office of MDPI; we have attached the English-editing certificate-47892 with this manuscript submission.
- Reviewer 1 comment: The manuscript needs to be checked for statistical significance and indicated with **, especially in fig 3,4.
Author response: According to Reviewer 1 request, Figures 2 and 3 have been modified using asterisks instead of the p values. The corresponding figure legends were modified for better understanding also. Besides, we have double checked data and statistical significance showed in the first version of the manuscript and they are correct. In all cases, and according to every statistical test requirement, we considered every corresponding assumption test; such as, whether standard deviations were equal or if data sampled from Gaussian distributions, among others.
Regards Figure 4, because the semiquantitative nature of the experiment (optical densities obtained from zymograms) there is not enough data to perform an statistical test, since the data come from a couple of experiments. For that reason, and to avoid any confusion, we have removed error bars from the clustered bar chart, letting know to the reader the proportional changes observed in the MMP-2 and -9 expression after stimulation of fibroblast and macrophage co-cultures. As support for this argument, we are sharing with this response the reference of a paper previously published by us with the same situation[Salgado RM, Cruz-Castañeda O, Elizondo-Vázquez F, Pat L, De la Garza A, Cano-Colín S, Baena-Ocampo L, Krötzsch E. Maltodextrin/ascorbic acid stimulates wound closure by increasing collagen turnover and TGF-β1 expression in vitro and changing the stage of inflammation from chronic to acute in vivo. J Tissue Viability. 2017 May;26(2):131-137. doi: 10.1016/j.jtv.2017.01.004].
- Reviewer 1 comment: The introduction needs to be concise.
Author response: According to Reviewer 1 suggestion, two paragraphs from the Introduction section were removed, since they were redundant or lack of background relevance.
- Reviewer 1 comment: Abstract and conclusion should include a sentence proposing the future direction of the present research. Also the commercial aspects of how pharma or biotech industries can benefit.
Author response: According Reviewer 1 request, we have improved Abstract and Conclusion sections with some information regards future directions of this work, as well as commercial importance of the synthetic materials vs. the traditional gauze. See page 1, lines 24-26; page 10, lines 389-396.
- Reviewer 1 comment: Most of the methodology lacks citing the references from where the technique was adopted and used to conduct the study.
Author response: References for previously methodologies lack of citing have been included. See page 2, lines 77 and 78; page 3, line 109; page 4, line 172.
- Reviewer 1 comment: Below are a few suggestions the authors are requested to discuss and cite in appropriate sections.
Author response: We appreciate Reviewer 1 suggestion and references. We have included relevant information from some of those papers. See Introduction page 2, lines 51 and 52; Discussion page 9, lines 309-315.

Reviewer 2 Report
Authors Silvestre et al. demonstrated the in vitro the direct effect of a dialkyl carbamoyl chloride (DACC)-coated dressing on Staphylococcus aureus adhesion and growth in the article entitled “Dialkyl carbamoyl chloride–coated dressing prevents macro- phage and fibroblast stimulation via control of bacterial growth: An in vitro assay”. The DACC-coated dressing showed bacteriostatic properties and bacterial adsorption, enabling the avoidance of wound inflammation exacerbation. The work is interesting and statistical analyses are convincing thus can be acceptable in MDPI microorganisms. However, the paper should be revised carefully before accepting for publication.
Few comments are given below:
- Some of the long sentences are written in multiple sections, especially in abstract are to reader-friendly. Thus, it is recommended to break down the sentences to make it more reader-friendly.
- In most cases, the abbreviations are used without prior illustration such as CFU, ELISA. Please check carefully the whole manuscript.
- The language, grammatical and typo errors should be checked carefully throughout the manuscript.
- Not sure, why authors underlined the S. aureus throughout the manuscript.
- Authors are recommended to add more magnified SEM micrograph for confirming shape morphology and biofilm of S. aureus either in the Fig. 1 or supplementary.
- Conclusions are incomplete. Please summarize all of the important results and explain the significance of results in the Conclusion section.
Author Response
Microorganisms Journal
Reviewer 2
Dear Reviewer 2:
Please find the revised version of the manuscript entitled “Dialkyl carbamoyl chloride–coated dressing prevents macrophage and fibroblast stimulation via control of bacterial growth: An in vitro assay”. Manuscript ID: microorganisms-1809877.
In this new version of the work we have significantly improved the manuscript. We have included one supplementary material, and we have modified some figures in order to be more explicit. Awkward sentences have been revised and corrected and the manuscript has been reviewed by the language editing office of MDPI; we have attached the English-editing certificate-47892 with this manuscript submission. In general, we consider that reviewers’ comments helped us to improve the work in a final clear and purposeful version. Below you can find a point by point response, where every change of the manuscript is tracked according to the corresponding line(s). Since the manuscript has been rewritten we have not tracked changes on it.
Respectfully,
The correspondence author.
Reviewer 2:
- Reviewer 2 comment: Some of the long sentences are written in multiple sections, especially in abstract are to reader-friendly. Thus, it is recommended to break down the sentences to make it more reader-friendly.
Author response: According to Reviewer 2, in the new version of the manuscript we have improved the manuscript and it has been reviewed by the language editing office of MDPI; we have attached the English-editing certificate-47892 with this manuscript submission.
- Reviewer 2 comment: In most cases, the abbreviations are used without prior illustration such as CFU, ELISA. Please check carefully the whole manuscript.
Author response: In this new version of the manuscript we have double checked abbreviations in the whole manuscript and we have written the corresponding meaning.
- Reviewer 2 comment: The language, grammatical and typo errors should be checked carefully throughout the manuscript.
Author response: According to Reviewer 2, in the new version of the manuscript we have improved the manuscript and it has been reviewed by the language editing office of MDPI; we have attached the English-editing certificate-47892 with this manuscript submission.
- Reviewer 2 comment: Not sure, why authors underlined the aureus throughout the manuscript.
Author response: Usually when a word is written in another language than the language of the manuscript, the word must be highlighted by italics. However when the word is on italics per se, the word must be underlined to show that it was written in another language.
- Reviewer 2 comment: Authors are recommended to add more magnified SEM micrograph for confirming shape morphology and biofilm of aureus either in the Fig. 1 or supplementary.
Author response: According to Reviewer 2 suggestion, we have included an additional supplementary figure (Figure 1S in this new version of the manuscript) with a higher magnification of SEM images where cell shape and biofilm can be confirmed. Also, we have included in the present version of the manuscript the corresponding sentence in the results section. See page 4, lines 196-198.
- Reviewer 2 comment: Conclusions are incomplete. Please summarize all of the important results and explain the significance of results in the Conclusion section.
Author response: In the new version of the manuscript the Conclusion section has been improved according to Reviewers suggestions.

Reviewer 3 Report
The manuscript entitled “Dialkyl carbamoyl chloride–coated dressing prevents macrophage and fibroblast stimulation via control of bacterial growth: An in vitro assay” can not be accepted for publication in Microorganisms.
The evidence for DACC-coated dressings in preventing and treating infection without adverse effects was reviewed by Totty et al., 2017 (doi: 10.12968/jowc.2017.26.3.107) that analysed 17 studies with a total of 3408 patients observed. So, the idea of Ortega’s article is not new.
On the other hand, many technical problems need to be solved. Why were the macrophages treated with bacterial supernatant at a concentration of 4.5% while the co-cultures were treated with 50% bacterial supernatant? How were established these concentrations? What concentrations were used in the evaluation of cytotoxicity?
Why was TGF-b evaluated only in the co-culture experiment, while TNF-a was evaluated in both macrophages and co-culture?
What are the differences between control and basal medium? It is not clear.
The authors have to change some inappropriate text formulations.
For example:
-line 124 “1 mL supplemented DMEM with antibiotics”. Does this means complete medium (as in lines 87-89: “Dulbecco's modified Eagle medium (DMEM; GibcoTM, Life Technologies, Brooklyn, NY, USA) supplemented with 10% fetal bovine serum (GibcoTM) and 2 mM glutamine (GibcoTM)”? What antibiotic was used in the cells maintaining media? What was antibiotic concentration used in the experiment “Stimulation of eukaryotic cells with supernatants of S. aureus cultures.”?
However, the English need to be thoroughly improved.
Author Response
Microorganisms Journal
Reviewer 3
Dear Reviewer 3:
Please find the revised version of the manuscript entitled “Dialkyl carbamoyl chloride–coated dressing prevents macrophage and fibroblast stimulation via control of bacterial growth: An in vitro assay”. Manuscript ID: microorganisms-1809877.
In this new version of the work we have significantly improved the manuscript. We have included one supplementary material, and we have modified some figures in order to be more explicit. Awkward sentences have been revised and corrected and the manuscript has been reviewed by the language editing office of MDPI; we have attached the English-editing certificate-47892 with this manuscript submission. In general, we consider that reviewers’ comments helped us to improve the work in a final clear and purposeful version. Below you can find a point by point response, where every change of the manuscript is tracked according to the corresponding line(s). Since the manuscript has been rewritten we have not tracked changes on it.
Respectfully,
The correspondence author.
Reviewer 3:
- Reviewer 3 comment: The evidence for DACC-coated dressings in preventing and treating infection without adverse effects was reviewed by Totty et al., 2017 (doi: 10.12968/jowc.2017.26.3.107) that analysed 17 studies with a total of 3408 patients observed. So, the idea of Ortega’s article is not new.
Author response: With respect for the Reviewer 3, data emerged from the paper published by Totty et al. did not support the idea proposed by our group. Reasons to justify that comment are the following:
- First, Totty’s work is an attempt to perform a meta-analysis from clinical papers regards the use of the DACC-coated dressing in wounds. Our work is an in vitro trial, where we have demonstrated, where and how many bacteria, were distributed in a closed system, such as the dressing cultured in a 24-well culture plate. Here, we have quantitated the CFU from the supernatants, dressings and the residual biofilms.
- Second, nor Totty’s work or another paper under our knowledge, have demonstrated, in vitro or in vivo like us, that the use of the DACC-coated dressing can adhere bacteria ( aureus in this case) avoiding bacteriolysis; at least indirectly by our experiments. For that reason, we have been cautious and we included in the first version of the manuscript the sentence “suggesting that the physical removal of bacteria has advantages over their killing by antibiotic or antiseptic means, due to the control of excessive local inflammation and remodeling” (Discussion section of the new version of the manuscript, page 9, lines 356-358).
- We considered that the Totty’s work is written carelessly. For example, Totty JP and co-workers established a series of ideas that are not supported by the bibliography they have included:
- In the Introduction section of the Totty’s paper, they established “…and will therefore irreversibly adhere to the DACC coating on dressings.5”. Reference 5 (Ljungh Å, Wadström T. Growth conditions influence expression of cell surface hydrophobicity of staphylococci and other wound infection pathogens. Microbiol Immunol 1995;39(10):753–7) did not mention anything about the reversibility of bacterial attachment to the DACC-coated dressing.
- Also, in the same section, Totty et al. mention that “Subsequent dressing changes will then result in the removal of large numbers of microbes and a decreased bacterial load at the wound site.6”. Reference 6 (Ljungh Å, Yanagisawa N, Wadström T. Using the principle of hydrophobic interaction to bind and remove wound bacteria. J Wound Care 2006;15(4):175–80) did not mention anything about the proportion of microorganisms removed from the wound site, data came from in vitro assays and reported in the review from Ljungh Å et al.
- Once again, in the Introduction section, Totty et al. mention that “Perhaps most importantly, since the mechanism of antibacterial action is of physical binding and removal, there is no risk of bacteria developing resistance, and the lack of bacteriolysis prevents endotoxin release to the wound bed.8” . Reference 8 (Probst A, Norris R, Cutting K. Cutimed® Sorbact® Made easy. Wounds International 2012;3(2):1–6) is a pseudopaper, published under the “journal” format, but in fact is a commercial publicity about the DACC coated dressing. Indeed, that information never supports bacteriolysis prevention by the clinical or in vitro use of the dressing.
- On the other hand, Totty JP and co-workers established that “No meta-analysis of trial data was possible for the included studies, due to differences in trial methodology and outcome measures.”. It means, their review did not demonstrate anything about the useful of the DACC-coated dressing.
- Finally, Totty JP and co-workers concluded “In general, the outcomes from these studies are positive; however, many of the outcome measures were highly subjective”.
Several works have demonstrated how the DACC-coated dressing attaches bacteria by descriptive analyses (i.e. SEM) or quantitative assays (CFU or fluorescence quantification). However, in our knowledge, we are the first that quantitate S. aureus CFU from the different environments of the culture, as well as the first to report how the filtered supernatants can modulate eukaryotic cell behavior.
- Reviewer 3 comment: On the other hand, many technical problems need to be solved. Why were the macrophages treated with bacterial supernatant at a concentration of 4.5% while the co-cultures were treated with 50% bacterial supernatant?
Author response: To explain this experimental strategy we considered some environmental aspects:
- Regards to eukaryotic stimulation by filtered supernatants in the first part of our work, we evaluated the minimum proportion of bacterial supernatant capable to stimulate TNF-α release from macrophages treated with oxacillin, mainly because the supernatants became from bacteria growth in Müller-Hinton broth that is not an appropriate medium for eukaryotic cells. Previously we had evaluated a final 10-fold dilution of the supernatants as previously assayed by van Langevelde P et al. (van Langevelde P, van Dissel JT, Ravensbergen E, Appelmelk BJ, Schrijver IA, Groeneveld PH. Antibiotic-induced release of lipoteichoic acid and peptidoglycan from Staphylococcus aureus: quantitative measurements and biological reactivities. Antimicrob Agents Chemother. 1998, 42, 3073-8. doi: 10.1128/AAC.42.12.3073.
- Macrophages were chosen because different works have studied macrophage cytokine (TNF-α) release as an indicator for bacteriolysis. In such manner that in the first part of the work we could demonstrate that supernatants from aureus planktonic cultures treated in vitro with the DACC-coated dressing prevent macrophage activation.
- Then, we wondered, whether a culture medium with biochemical properties “similar” to the wound exudate (culture medium containing fetal bovine serum), also was capable to favor aureus attachment to the DACC-coated dressing and prevent bacteriolysis. The 50% bacterial supernatant definition was due to the results obtained in the experiments of bacterial behavior, where bacterial cultures grown in DMEM 10% FBS exhibited lower S. aureus replication than in Müller-Hinton broth; so, it would be necessary to use enough supplemented DMEM supernatants capable to stimulate eukaryotic cells. Also, we considered that, since the supernatants became from the same culture medium (without antibiotic) where eukaryotic cells grow, then we only needed to refresh the culture medium with the other 50% of DMEM 10% FBS with antibiotics and we could obtain enough macrophage activation.
- Reviewer 3 comment: How were established these concentrations?
Author response: Please see above (Author response 2 for Reviewer 3).
- Reviewer 3 comment: What concentrations were used in the evaluation of cytotoxicity?
- Author response: For the cytotoxicity assay we used 50% bacterial supernatant concentration. In the first version of the manuscript we referred it in the section 2.3 Cytotoxicity assay, under the following sentence “…we prepared and treated macrophage cultures and fibroblast/macrophage co-cultures as described previously.”, and previously (section 2.2 Stimulation of eukaryotic cells with supernatants of aureus cultures) we had defined the 50% bacterial supernatant concentration as follows: “…and the co-cultures were treated with a mixture of 125 μL fresh medium and 125 μL filtered supernatant from S. aureus cultures incubated with and without dressing”. (Subsection 2.3 Cytotoxicity assay of the new version of the manuscript, page 3, lines 138-140 and Subsection 2.2 Stimulation of eukaryotic cells with supernatants of S. aureus cultures of the new version of the manuscript, page 3, lines 130-132).
- Reviewer 3 comment: Why was TGF-β evaluated only in the co-culture experiment, while TNF-α was evaluated in both macrophages and co-culture?
Author response: As we have established before, Author response 2 for Reviewer 3 comment, the first intention was to evaluate the effect of supernatants from S. aureus cultures on macrophage activation (TNF-α release), in order to know indirectly the lack of bacteriolysis by the treatment with the DACC-coated dressing. However, when we assessed supernatants from S. aureus cultures grown in a culture medium with “similar” characteristics of the wound exudate (DMEM 10% FBS), we considered important to perform the evaluation in a biological environment closer than a wound also, where besides macrophages, fibroblasts are the most abundant cells in the wound bed. Even more, the macrophage and fibroblast co-culture would represent a paracrine condition observed in any wound, so it would be necessary to evaluate the indirect bacteriolysis through TNF-α release, but also a representative growth factor from those cells that participate in the wound repair process and inflammation. So, we choose TGF-β.
- Reviewer 3 comment: What are the differences between control and basal medium? It is not clear.
Author response: As stated in figure legends (Figure 3, 4 and 1S), “the basal term refers to the conditioned medium from unstimulated eukaryotic cultures”. It means that the experiments performed in eukaryotic cell cultures needed two controls. The first was the control we had been performing the whole study, the supernatants from the bacterial growth without any treatment. And the second, a control regards the growth of eukaryotic cells without any treatment. Since two controls would be confuse, we named the second one as “basal”, because it is the eukaryotic cell behavior under basal conditions (DMEM 10% FBS).
- Reviewer 3 comment: The authors have to change some inappropriate text formulations. For example:
-line 124 “1 mL supplemented DMEM with antibiotics”. Does this means complete medium (as in lines 87-89: “Dulbecco's modified Eagle medium (DMEM; GibcoTM, Life Technologies, Brooklyn, NY, USA) supplemented with 10% fetal bovine serum (GibcoTM) and 2 mM glutamine (GibcoTM)”?
Author response: According to Reviewer 3 request we have revised the whole methods and results in order to be explicit with the formulations. In a series of experiments performed in this work we used DMEM as culture medium for S. aureus growth. DMEM was supplemented with 10% of fetal bovine serum and glutamine; this formula has been called “supplemented DMEM”. On the other hand, when DMEM was used for eukaryotic cell cultures, it was enhanced with antibiotics (penicillin/streptomycin). For that reason, the last formula has been called "supplemented DMEM with antibiotics".
- Reviewer 3 comment: What antibiotic was used in the cells maintaining media?
Author response: As was stated in the first version of the manuscript (Section 2.2 Stimulation of eukaryotic cells with supernatants of S. aureus cultures), the antibiotics used were 100 U/mL penicillin and 100 µg/mL streptomycin (a commercial formula from GibcoTM, Life Technologies, Brooklyn, NY, USA. (See page 3, lines 113 and 114 of the new version of the manuscript).
- Reviewer 3 comment: What was antibiotic concentration used in the experiment “Stimulation of eukaryotic cells with supernatants of S. aureus cultures.”?
Author response: Please see above (Author response 8 for Reviewer 3).
- Reviewer 3 comment: However, the English need to be thoroughly improved.
Author response: According to Reviewer 3, in the new version of the manuscript we have improved the manuscript and it has been reviewed by the language editing office of MDPI; we have attached the English-editing certificate-47892 with this manuscript submission.

Round 2
Reviewer 2 Report
The authors addressed all of the suggested corrections thus the revised version of the manuscript can be accepted
Reviewer 3 Report
The Authors of the manuscript “Dialkyl carbamoyl chloride–coated dressing prevents macrophage and fibroblast stimulation via control of bacterial growth: An in vitro assay” made some improvements in the revised form of their paper. However, I’m still not convinced that this is paper is suitable for publication.
The idea of using the DACC-coated dressing in bacterial infections is not new, as the authors said:“Several works have demonstrated how the DACC-coated dressing attaches bacteria by descriptive analyses (i.e. SEM) or quantitative assays (CFU or fluorescence quantification)”. Also, “how the filtered supernatants can modulate eukaryotic cell behavior” is an idea quite intensely debated. Putting these ideas together is very good, but the problems of this article are regarding how experiments were conducted. In order to demonstrate that “supernatants from aureus planktonic cultures treated in vitro with the DACC-coated dressing prevent macrophage activation” the authors used in their experiments “the minimum proportion of bacterial supernatant capable to stimulate TNF-α release from macrophages treated with oxacillin” due to Müller-Hinton broth cytotoxicity. Have you thought if the results obtained are not due to the too low concentration tested? In another experiment, it was used a culture medium containing fetal bovine serum, “a culture medium with biochemical properties <similar> to the wound exudate”, at a concentration “capable to stimulate eukaryotic cells”. Why was this supernatant not used in all experiments? These two culture media and the supernatants of bacteria grown in these media induce different effects in eukaryotic cells, and it is incorrect to compare the results. On the other hand, the concentration of oxacillin used is not clear. Was it the same in all experiments or not?